# Reconstructing, Understanding, and Analyzing Relief-Type Cultural Heritage from a Single Old Photo

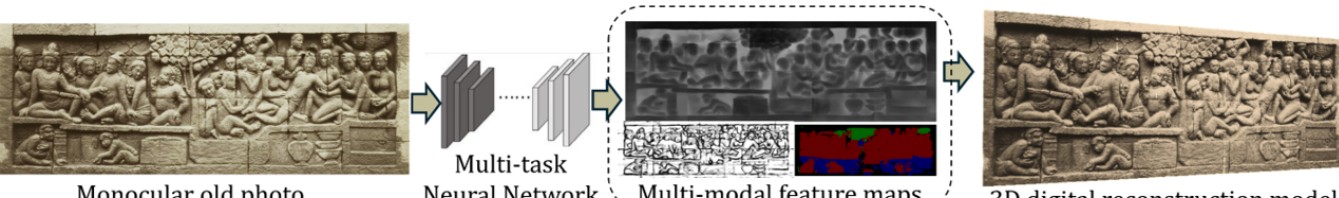

**Figure 1: We propose a multi-task neural network to predict multi-modal feature maps including depth, semantics and edges from a single old photo. The proposed method not only enables monocular 3D digital reconstruction of damaged or lost relief-type cultural heritage objects, but also improves understanding and analysis towards the relief scenario.**

## ABSTRACT

Relief-type cultural heritage objects are commonly found at historical sites but often manifest with varying degrees of damage and deterioration. The traditional process of reconstructing these reliefs is laborious and requires extensive manual intervention and specialized archaeological knowledge. By utilizing a single old photo containing predamage information of a given relief, monocular depth estimation can be used to reconstruct 3D digital models. However, extracting depth variations along the edges is challenging in relief scenario due to the highly compression of the depth values, resulting in low-curvature edges. This paper proposes an innovative solution that leverages a multi-task neural network to enhance the depth estimation task by integrating the edge detection and semantic segmentation tasks. We redefine edge detection of relief data as a multi-class classification task rather than a typical binary classification task. In this paper, an edge matching module that performs this novel task is proposed to refine depth estimations specifically for edge regions. The proposed approach achieves better depth estimation results with finer details along the edge region. Additionally, the semantic and edge outputs provide a comprehensive reference for multi-modal understanding and analysis. This paper not only advances in computer vision task computer vision tasks but also provides effective technical support for the protection of relief-type cultural heritage objects.

## CCS CONCEPTS

• **Applied computing** → **Arts and humanities**; • **Computing methodologies** → **Multi-task learning**; **Neural networks**.

**Unpublished working draft. Not for distribution.**

## KEYWORDS

Relief, Cultural Heritage, Multi-task Learning, 3D reconstruction

**ACM Reference Format:**

. 2018. Reconstructing, Understanding, and Analyzing Relief-Type Cultural Heritage from a Single Old Photo. In *Proceedings of Make sure to enter the correct conference title from your rights confirmation emai (Conference acronym 'XX)*. ACM, New York, NY, USA, 10 pages. https://doi.org/XXXXXXX.XXXXXXX

## 1 INTRODUCTION

Relief-type cultural heritage objects, commonly found at historical sites worldwide, are important in humanistic areas such as art, culture, history, and architecture. Unfortunately, these valuable cultural heritage objects often suffer from varying degrees of damage and loss caused by natural or human factors [54]. While scanning and photogrammetry technology can achieve permanent three-dimensional (3D) digital preservation of their current states, they cannot restore their appearance prior to damage and loss [60]. The traditional process of restoring these reliefs is laborious, requiring extensive manual intervention and specialized archaeological knowledge [24, 56]. Moreover, reconstruction methods for typical 3D cultural relics often involve deep learning-based point cloud completion processes to reconstruct damaged and missing structures. However, these methods are generally suitable only for scenes with simple structures and minimal damage [16, 44, 47]. Unfortunately, these may not be applicable to relief-type cultural heritage objects, which are typically complex in their geometric structure and extensive in scale and quantity.

Fortunately, many of these precious reliefs are often documented in old photos, as shown in Figure 2. These 2D monocular old photos can serve as very effective references for the 3D reconstruction of relief-type cultural heritage objects. First, relief scenes are different from 3D scenes such as sculptures or 2D scenes such as paintings, and are more like special 2.5D scenes. Second, the reliefs are usually only meant to be viewed from the front or side. Therefore, these two characteristics allow a single image taken, from the front, to effectively cover most of the content, providing sufficient information for monocular algorithms.

(a)        (b)        (c)

**Figure 2: Examples of damaged or lost relief: (a) Juyong Pass, partially damaged [3]; (b) Yungang Grottoes, partially damaged [66]; (c) Borobudur Temple, overall lost [39]. Old photos with predamage information, and photos of the present situation, are presented on the top and bottom, respectively.**

Pan et al. first proposed a 3D reconstruction method based on monocular depth estimation to reconstruct buried reliefs in the Borobudur temple [31]. Although they achieved 95% in reconstruction accuracy, their result lack detailed fine structures of the carved items, e.g., human figure facial features and many kinds of decorations. This limitation is due to the inadequate extraction of depth variations in the edge regions that form the fine structures. In a relief scenario, the depth values are significantly compressed compared to other 3D scenarios, leading to less pronounced changes along the edges. These edges cannot be easily detected from 2D images, as discussed in Section 2. Pan et al. identified this characteristic as "soft edge" and extracted this unique information based on the curvature changes in 3D space [20]. Subsequently, a soft edge map was straightforwardly input into their network as auxiliary information [32]. Their experimental results demonstrated that soft edge information effectively improves the depth estimation task in relief data. However, from our perspective, there are three critical limitations that still need to be addressed:

(1) The edge map is extracted from 3D points and used as an additional input, which significantly reduces the accuracy of their proposed model on the test data. This decrease in accuracy occurs because, during the testing phase, 3D points from an old photo are unavailable, thereby leaving no effective method to obtain an accurate edge map.

(2) The soft edge map is incorporated into the network in the later part of the decoding stage, which limits its impact on the depth estimation task.

(3) They represent the soft edge map as a binary image to distinguish between edge and non-edge areas as a mask for further processing. However, the binary representation fails to convey the 3D curvature changes inherent in soft edges, which could provide effective clues for depth estimation task.

In this paper, we propose solutions to the aforementioned limitations using a multi-task neural network equipped with a novel edge matching module that performs a newly defined soft edge extraction task. For the first limitation, the proposed method, based on multi-task learning (MTL), is capable of performing both soft edge detection and depth estimation tasks within the same neural network. Upon proper training, this neural network can provide accurate edge information for test data through its edge detector. To address the second limitation, we design a novel edge matching module that directs the depth estimation task to focus more intensively on the soft edge region, thereby yielding more detailed depth estimation result. For the third limitation, we redefine soft edge detection as a multi-class classification problem, aimed at determining the degree of "softness" of specific edge regions. This approach preserves the crucial curvature information, thus enhancing the effectiveness of the depth estimation task.

Following the work of Pan et al., we introduce one of the UNESCO World Heritage Sites, Borobudur Temple in Indonesia, as our experimental subject. The temple features 2,672 bas-relief panels on its walls, constituting the largest collection of Buddhist reliefs in the world. Unfortunately, due to safety concerns, the temple's foot encasement was reinstalled, and 156 relief panels carved into this encasement were obscured by stone walls and are now hidden and invisible. For each of these hidden panels, a grayscale photograph taken in 1890 is preserved (refer to Figure 2) [5, 39]. We digitize the visible reliefs into 3D models and establish a training dataset including RGB inputs, depth, semantics, and soft edge labels to train our proposed model. The results of our quantitative and qualitative experiments demonstrate that our method provides a more detailed depth map than Pan et al. and other state-of-the-art (SOTA) depth estimation models, with richer detail in the edge regions, leading to superior 3D digital reconstruction models. Furthermore, our proposed multi-task neural network also delivers semantic segmentation and soft edge extraction results, achieving performance comparable to SOTA models and enhancing multi-modal understanding and analysis of relief data.

We summarize the contributions of this paper as follows: (1) We propose a novel multi-task network that enables the reconstruction, understanding, and analysis of relief-type cultural heritage from a monocular old photo; (2) We introduce a novel edge matching module within the network that performs a newly defined soft edge detection task, enhancing the details in the edge regions of depth estimation results and thus enabling the reconstruction of more accurate 3D digital models; (3) We propose a dynamic edge-enhanced loss function to optimize the proposed neural network; (4) We apply this method to reconstruct the Borobudur hidden reliefs their remaining old photos, thereby aiding in the preservation of this invaluable cultural heritage site.

## 2 RELATED WORK

**Relief Reconstruction and Generation:** The traditional process of reconstructing relief-type cultural heritage objects into 3D digital models is laborious [24, 56]. Deep learning-based reconstruction of relief objects has rarely been studied, Pan et al. first proposed a monocular depth estimation-based approach to reconstruct the relief from a single monocular photo[31], and improved the results by involving soft edges in their later work[32]. However, there are still limitations that remain to be addressed, as we describe in Section 1. However, deep learning-based reconstruction is a relatively common for other 3D relics. With scanned data of intact objects, it is possible to reconstruct cultural heritage objects by point generation approaches to fill the missing structures [16, 44, 47]. However, at present, these methods are only suitable for small

areas of broken artifacts with simple geometries, thus they are not very versatile for relief scenes.

Additionally, there are related studies on relief generation task (including noncultural heritage objects). Some use 3D models for depth compression to generate reliefs that do not meet the standards for the reconstruction of relief-type cultural heritages from old photos [18, 19, 63]. However, it is worth mentioning that the effect of edge information in relief has also been proven in these works. JI et al [19] demonstrated that hierarchical relationships change dramatically with a gradual decrease in depth, resulting in continuous degradations of the original details. The proposed edge optimizations in their work improved the model-to-relief reconstruction results. Other image-based methods generally rely on other prior inputs [28, 37, 43] or human-computer interactions (HCIs) [61]. Moreover, these studies are generally optimized for a few specific simple characters, such as plants, human faces or other objects with similar geometric structures[64].

**Monocular Depth Estimation:** Deep learning-based methods have become the mainstream solution for monocular depth estimation task because they effectively learn depth representations in an end-to-end manner. Eigen et al. proposed the first depth estimation network, which is a multi-scale fusion network, to regress the depth value [13]. Following this, considerable improvements have been made by utilizing or modifying superior network backbones [1, 15, 25], carefully designing the regression task as a classification task [6, 27], introducing more priors [35, 36, 52] and better objective functions [53, 62]. However, despite their promising performance, they are difficult to generalize to unseen domains, especially in relief scenes with unique data features.

Recently, several works apply diffusion model-based methods to monocular depth estimation [12, 40]. By taking advantage of conditional diffusion models [21], zero-shot depth estimation has made breakthroughs [58]. However, these models require large amounts of training data at their initial stage. Marigold is specially designed for tuning with small amounts of data, but 74k is still needed [21]. As data collection of cultural heritage objects is often limited by protection purposes or data security, no large public relief dataset is currently available. Moreover, as shown in our experimental results, the details in depth result cannot be estimated properly by these models. To address these demands, our research approach is to preparing various labels on very few sample data to perform related tasks with depth estimation based on effective MTL architecture.

**Edge Detection:** Different from contours and boundaries, which correlate with semantically meaningful entities, edge detection aims to capture all significant intensity discontinuities in an image [67]. Traditional edge detectors [7, 23], learning-based methods [11, 29], convolutional neural networks [33, 46], and recent approaches utilizing vision Transformers [34] have been explored for edge detection in 2D images. These methods focus on identifying notable brightness changes and classify pixels into two categories: edges and non-edges. However, as discussed in Section 1, soft edges, which are more indicative of 3D curvature changes, cannot be adequately represented by a binary classification. Kawakami et al.[20] extracted soft edges in reliefs using a 3D feature highlighting method that employs point opacity provided by a transparent rendering technique[49, 51]. Subsequently, Pan et al. [32] project these 3D soft edges onto a binary edge map and utilized it as an additional input for their depth estimation network. To further leverage soft edge information, we project 3D soft edges onto multi-class edge maps and use these as labels to train a deep learning-based edge detector designed for a multi-class classification task, thereby defining a novel soft edge detection task.

**Multi-task Learning:** Multi-task learning (MTL) enhances task performance compared to independent single-task training by leveraging shared information and representations across tasks [8]. To improve MTL performance, numerous efforts have focused on designing encoder architectures [42, 50], predicting intermediate auxiliaries [57], and developing novel loss functions [10]. Several studies have demonstrated improved performance by jointly estimating depth and semantic segmentation [14, 48]. Ji et al. [17] further established the relationship between depth and semantics in relief scenes in their work on semantic segmentation. Inspired by these findings, we incorporate semantic segmentation into our MTL approach to jointly predict depth, semantics, and edges. Additionally, to enhance the interplay between the depth estimation and edge detection tasks, we have developed a novel edge matching module within our proposed MTL architecture.

# 3 METHOD

## 3.1 Overview

As shown in Figure 1, we propose a multi-task neural network to extract various features involving the information of depth, semantics, and edges from a single old photo of a given relief-type cultural heritage object. These three kinds of feature maps provide the multi-modal data foundation for understanding and analysis of the relief scenario. Moreover, by predicting a corresponding dense depth map from a single old photo, the 3D digital model can be reconstructed. This effectively addresses the widespread issue of varying degrees of damage and loss in relief-type cultural heritage objects, because the predamage information is often documented in their historical photographs.

The key to improving the depth estimation results of relief data lies in extracting more information on subtle depth variations at the edge regions. Utilizing edge images to effectively assist in the depth estimation task is a highly effective method. As we discussed in 1 Introduction, Pan et al. [32] has made preliminary attempts, but there is still significant room for improvement. Therefore, we propose a novel method based on multi-task learning to address this issue.

First, the soft edge extraction task is newly defined in Section 3.2 to better extract and represent the data features of the relief-type cultural heritage objects. Second, a multi-task neural network with novel architecture is proposed in Section 3.3 to effectively utilize the soft edge information. The network can extract the depth, semantic, and edge features, and further constrain the depth estimation task to focus on the soft edge regions using a proposed edge matching module. Finally, the dynamic edge enhanced loss function is proposed in Section 3.4, so that the proposed network can be better optimized.

## 3.2 Soft edge detection task

As shown in Figure 3, traditional edge detection tasks treat the detection of 2D edges as a binary classification problem, segregating each pixel into either edge or non-edge regions. In public datasets, the differential brightness within edge regions reflects the degree of consensus among humans during the manual labeling process [30]. This method suffices for depicting brightness transitions along edges in conventional scenes. However, the edges in relief data signify not merely changes in brightness but variations in curvature change across 3D space.

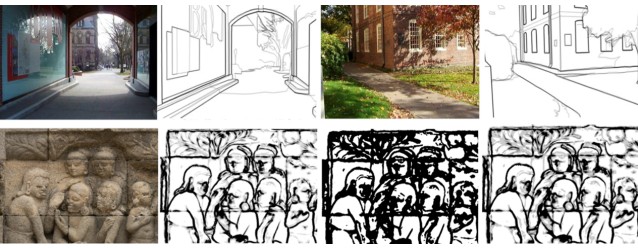

**Figure 3: Edge Maps in the open source dataset MDBD (Top). Variations in edge map representations of relief data (Bottom), from left to right: continuous representation, binary representation, and the proposed multi-class representation.**

Therefore, we propose to define the edge detection of relief data as a multi-classification task to better express this important feature. As shown in Figure 3, by employing a 3D edge highlighting technique [20], soft edge regions can be detected and subsequently projected onto a edge map. This map features continuous pixel values ranging from 0 to 255. Unlike related studies that directly binaries the map for use as a mask, we categorize the pixels into six distinct classes based on their values as detailed in Table 1. For a given class $n$, the pixel value is determined by the formula $51 \times n$. This task is executed by the proposed soft edge detector (see Section 3.3), which is optimized by a meticulously refined loss function (see Equation 10).

**Table 1: division rules of each class.**

| Range | 0−49 | 50−99 | 100−149 | 150−199 | 200−249 | 250−255 |
|-------|------|-------|---------|---------|---------|---------|
| Class | 0 | 1 | 2 | 3 | 4 | 5 |
| Value | 0 | 51 | 102 | 153 | 204 | 255 |

## 3.3 Multi-task neural network

**Overall structure:** The network structure of our proposed method is shown in Figure 4. The network follows an encoder-decoder design to perform monocular depth estimation task and semantic segmentation task, and is additionally constrained by a novel edge matching module, which performs the soft edge detection task proposed in Section 3.2. The encoder is composed of a modified SwinV2 Transformer encoder following EMSAFormer [14] and a heavyweight VIT encoder following SAM [22]. Please note that the parameters of the SAM encoder are frozen in this work because our dataset is not sufficient for overall fine-tuning. We utilize the SAM

encoder to leverage its zero-shot feature extraction capability to obtain rich features. To merge the extracted features from the two independent encoders, two patch merging layers are set behind the SAM encoder to adjust the resolution and the number of channels of the intermediate results. Then, the adjusted features are fused through cross-attention to the features obtained from the SwinV2 Transformer encoder, followed by a context module. Moreover, triple skip connections are used to retain the low-level features and input to the two decoders. Additionally, the SwinV2 Transformer encoder incorporates an extra input edge image, which is output by the proposed soft edge detector of the edge matching module. The specific information about the edge matching module, soft edge detector, and two task decoders are as follows.

**Edge matching module:** As shown in Figure 4, the proposed module comprises two soft edge detectors designed to individually extract multi-class soft edge maps from the input monocular photo and the output depth map, respectively. Both detectors share identical structures, detailed in the subsequent paragraph. Moreover, the matching loss between the two produced soft edge maps is calculated in this module, which is part of the optimization object of the network following Equation 5. The logic behind this module is predicated on the notion that, only when the deep prediction results are sufficiently accurate can the edge information of details be extracted. This will lead to an increased similarity between the two edge result images, which in turn reduces the penalty on the optimization objective. This module effectively imposes a directed constraint on the depth prediction task, making it focus more on extracting the details outlined by the soft edges.

**Soft edge detector:** The proposed soft edge detector is modified from LDC [45], which is a lightweight network with just 674K parameters. As shown in (c) of Figure 4, the detector follows a CNN architecture with four intermediate edge maps (F1 to F4); hence, the final result comes from the fusion of these maps through four skip connections. The output soft edge map follows our proposed multi-classification representation, and the pixel values reflect the curvature change of the soft edges in 3D space. This lightweight backbone is selected because we are trying to complete a more complex edge detection task with a limited dataset. The parameters pretrained on the MDBD dataset [30] are utilized in this paper. Instead of making excessive adjustments to the detector structure, we propose a meticulously refined loss function to complete the newly defined task, following Equation 10 and Equation 11.

**Semantic decoder and depth decoder:** The decoders for the semantic segmentation task and depth estimation task are designed to suit the specific needs of each task. As shown in (a) and (b) of Figure 4, each decoder consists of a series of convolutional layers, batch normalization layers and upconvolutional layers. Moreover, the feature maps, with multiple scales generated by skip connections as we mentioned above, are fused with the intermediate results of both decoders via the concatenation operation. For semantic segmentation, the decoder projects to the number of semantic classes. For depth estimation, the decoder regresses continuous pixel values ranging from 0 to 255.

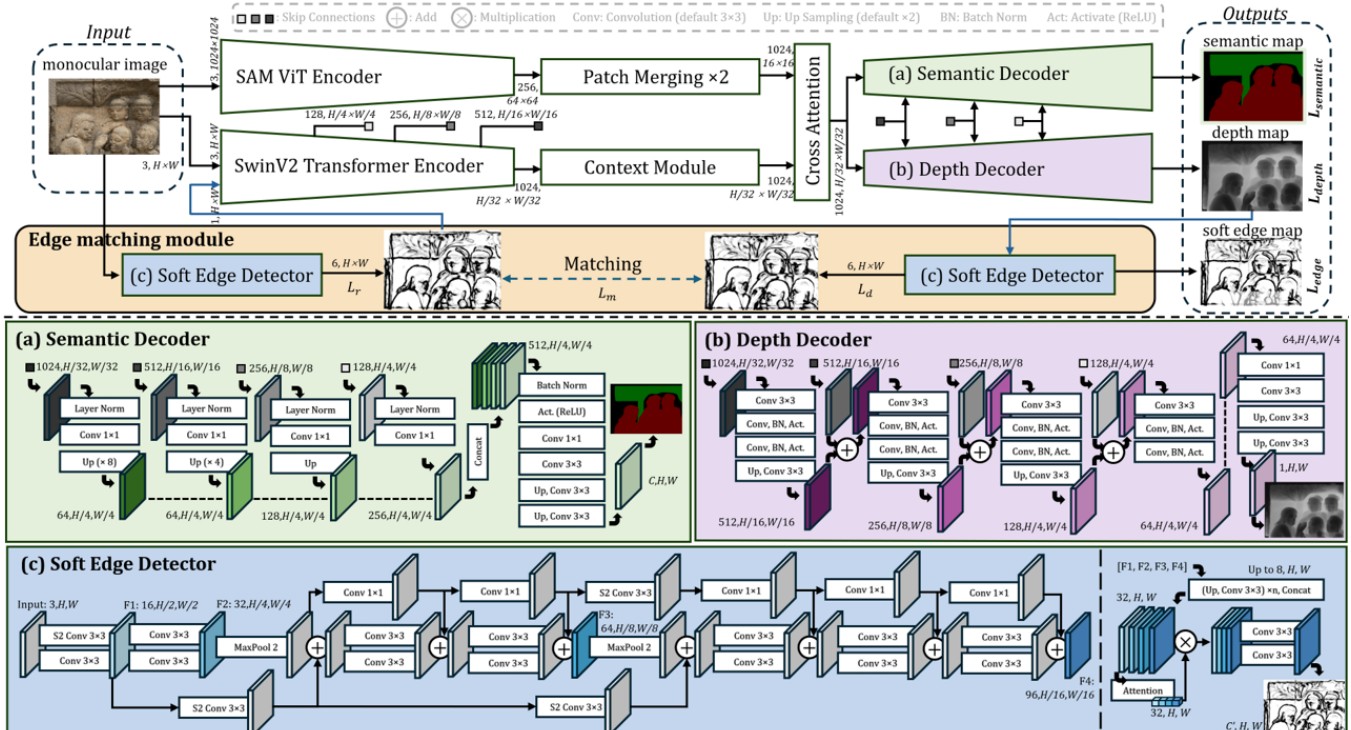

**Figure 4: Network structure of our proposed method. The top half shows the overall structure of the network, and the bottom half shows the details of the (a) semantic decoder, (b) depth decoder and (c) soft edge detector, respectively.**

## 3.4 Dynamic edge-enhanced loss function

The proposed multi-task learning network optimizes a novel dynamic edge-enhanced loss function. The total loss is joined by three tasks, depth estimation, soft edge detection and semantic segmentation, following Equation 1:

$$L = \alpha L_{semantic} + \beta L_{depth} + \gamma L_{edge} \tag{1}$$

where $L_{semantic}$, $L_{depth}$ and $L_{edge}$ denote the loss functions for the tasks of semantic segmentation, depth estimation and soft edge detection, respectively. $\alpha$, $\beta$ and $\gamma$, respectively, represent the weight coefficients for the three tasks.

For the semantic segmentation task and depth estimation task, we utilize the cross-entropy loss and the Silog loss [13] following the related works, respectively:

$$L_{semantic}(s', s) = -\frac{1}{n} \sum_{i=1}^{n} \sum_{k=1}^{c} p_k^{(i)} \log(q_k^{(i)}) \tag{2}$$

$$L_{depth}(d_e^{(i)}) = \frac{1}{n} \sum_{i=1}^{n} (d_e^{(i)})^2 - \frac{\lambda_d}{n^2} \left( \sum_{i=1}^{n} d_e^{(i)} \right)^2 \tag{3}$$

$$d_e^{(i)} = \log(d^{(i)}) - \log(d'^{(i)}) \tag{4}$$

where $n$ denotes the number of pixels with valid ground truth values, $c$ denotes the number of categories, which is set to 4, $s'$ denotes the label and $s$ denotes the predicted probability for each category after applying the softmax function. $d$ denotes the prediction result of depth estimation, and $d'$ denotes the ground truth of depth

estimation. $\lambda_d$ is set to 0.15 to be invariant to global-scale changes in the predicted depth map.

For the soft edge detection that we defined in this paper, a novel dynamic loss function is proposed. The total loss in this task should consider three terms: (a) the edge detection accuracy $L_r$ predicted from the input monocular image, (b) the edge detection accuracy $L_d$ predicted from the output depth map and (c) the matching loss $L_m$ of the two predicted soft edge maps. The total loss $L_{edge}$ of the soft edge detection task can be expressed as:

$$L_{edge} = \lambda_1 L_r + k(\lambda_2 L_d + \lambda_3 L_m) \tag{5}$$

$$L_r = l(e_r, e_g); \quad L_d = l(e_d, e_g); \tag{6}$$

$$L_m(e_r, e_d) = \frac{1}{n} \sum_{i=1}^{n} \left| e_r^{(i)} - e_d^{(i)} \right| \tag{7}$$

where $l$ is the loss function for the soft edge detector, with $e_r, e_d, e_g$ representing the edge maps detected from the input monocular image, the edge maps detected from the input monocular image, and the ground truth. $\lambda_1$, $\lambda_2$, and $\lambda_3$ are the weight coefficients for each loss term, and $k$ is the proposed dynamic control parameter. We first introduce the proposed dynamic control parameter $k$ and then explain the details of the loss function for the soft edge detection task $l$ as follows.

**Dynamic control parameter $k$:** During the initial training phase of the proposed network, the output depth map of the depth

decoder is generally poor and uncertain, from which meaningless soft edge detection results are produced. If these results are used to calculate $L_d$ and $L_m$, it will lead to instability $L_{edge}$ and total loss $L$, thereby affecting the learning effectiveness of each task and the entire network. Therefore, the dynamic coefficient $k$ is adopted to adjust the weights of $L_d$ and $L_m$ based on the accuracy of the depth estimation. This dynamic coefficient $k$ is set as follows:

$$k = 1 - \tanh^2(\lambda_k \hat{d}) \qquad (8)$$

$$\hat{d} = \frac{1}{n} \sum_{i=1}^{n} \left| d^{(i)} - d'^{(i)} \right| \qquad (9)$$

where $\hat{d}$ represents the accuracy of the depth estimation result using the $L^1$ norm, which fluctuates primarily within the range of 0.08 to 0.45. To ensure that the coefficient $k$ can exhibit a substantial dynamic range within the primarily active range of $\hat{d}$, the scaling coefficient $\lambda_k$ is set to $\lambda_k = 2.5$, and the tanh operator is adopted.

**Refined classification loss function $l$:** To effectively obtain the soft edge map defined in Section 3.2, we propose a refined classification loss function $l$ as follows:

$$l(e, e') = -\frac{1}{n} \sum_{i=1}^{n} \left[ \sum_{k=1}^{c'} e_k^{(i)} \log(e'^{(i)}_k) \right] + \lambda_e L^1(e, e') \qquad (10)$$

$$L^1(e, e') = \frac{1}{n} \sum_{i=1}^{n} \left| \arg\max(e^{(i)}) - \arg\max(e'^{(i)}) \right| \qquad (11)$$

where $e$ denotes the predicted result, $e'$ denotes the ground truth. $\lambda_e$ is a scaling factor, which is set to 0.2.

In the typical classification task, different categories are independent and unordered, with equal distances between each other. However, as shown in Table 1, the soft edge detection task that we define is based on ordered pixel values. For example, the difference in pixel values between Class 2 and Class 0 is less than the difference in pixel values between Class 5 and Class 0. However, there will be no difference when calculating loss using a typical cross-entropy loss function, which is the problem that we want to address. To make the penalty predicting Class 5 greater than the penalty predicting Class 2 for the given pixel of Class 0, $L^1$ function is added as a penalty term behind the typical cross-entropy loss.

## 4 EXPERIMENTAL RESULTS

### 4.1 Implementation Details

The proposed network was implemented in PyTorch and trained on a Quadro RTX 6000 GPU with 24 GB of GPU memory. The weights of the SAM ViT encoder were initialized by the ViT-B SAM model [22] (not trainable), while the weights of the SwinV2 Transformer encoder were pretrained on ImageNet. For training, we employed the Adam optimizer in combination with the OneCycleLR learning rate scheduler. This scheduler had a maximum learning rate of 0.01 and a percentage start of 0.2, spanning a total of 100 epochs. Our model was trained on the Borobudur relief dataset provided by Pan et al. [32], consisting of 6,424 patches cropped from only 11 images with $3072 \times 1024$ pixels. The total training duration exceeded 34 hours, with a batch size of 4 and an input size of $640 \times 512$ pixels. To avoid the risk of overfitting, we augmented the

images prior to feeding them into the network by applying Gaussian noise, performing random resizing and cropping, adjusting random HSV jitter, and applying horizontal flipping. For comparative experiments, we adopted the following parameter settings in the proposed loss function to achieve the best performance on depth estimation task. For Equation. 1, $\alpha$ is set to 1.0, $\beta$ to 3.0, and $\gamma$ to 5.0. For Equation. 5, we set $\lambda_1$ to 1.0, $\lambda_2$ to 0.5, and $\lambda_3$ to 0.05.

**Table 2: Quantitative results on depth and edge maps.**

| | Depth | | | | Edge (with different GT) | | |
|---|---|---|---|---|---|---|---|
| | RMSE↓ | RMSElog↓ | abs↓ | sq↓ | mIou↑ | Accuracy↑ | RMSE↓ |
| Exp 1 | 9.5321 | 0.4023 | 4.0038 | 1.9839 | - | - | - |
| Exp 2 | 9.4510 | 0.4246 | 3.8710 | 1.9429 | **0.6610** | **0.7978** | - |
| Exp 3 | **9.2762** | **0.4014** | **3.5941** | **1.8830** | 0.24779 | 0.5703 | **6.5810** |
| Exp 4 | 9.3647 | 0.4189 | 3.7464 | 1.9059 | - | - | 7.7890 |
| Exp 5 | 9.6645 | 0.3970 | 3.9379 | 2.0071 | 0.2402 | 0.5468 | 6.7043 |

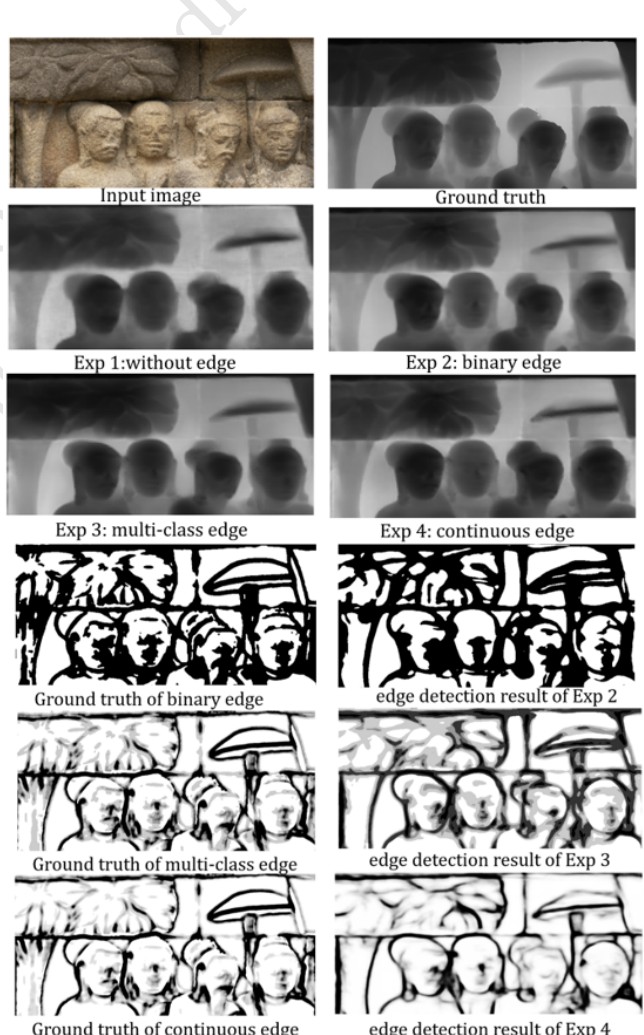

**Figure 5: Qualitative results on depth and edge maps.**

## 4.2 Ablation Study

In this section, the effectiveness of both the proposed edge matching module and conducting the soft detection task defined as a multi-class classification problem are evaluated. We demonstrate quantitative and qualitative experimental results on monocular depth estimation and edge detection when different settings of edge information are used in the proposed network. The results of the semantic segmentation were not significantly impacted by these settings. However, our proposed method still managed to achieve competitive results in the semantic segmentation task on relief data, details of which will be discussed in Section 4.3.

In our ablation studies, Exp 1 did not utilize the proposed edge matching module or any edge information. Exp 2 to Exp 5 all utilized the proposed edge matching module, each with different settings: Exp 2 involved binary edge, Exp 3 involved multi-class edge, Exp 4 involved continuous edge, and Exp 5 involved multi-class edge but without using the proposed dynamic control parameter $k$ in the proposed loss function as Equation 5.

As shown in Figure 5 and Table 5, the proposed edge matching module demonstrates the most significant improvement in extracting details from the edge regions. Compared to other experiments, Exp 1 displays the worst performance, with the highest error rates and the least detail in the edge regions. The newly defined soft edge detection task has also proven effective, as Exp 3 yields more accurate depth results in the edge regions than Exp 2, resulting in lower error metrics as indicated in Table 5. While Exp 2 did manage to extract some edge details, its depth value accuracy is inferior to that of Exp 3 when compared to the Ground Truth (GT). Please pay particular attention to the changes in depth values of the leaves and human ears in Figure 5. Additionally, the soft edge maps predicted in Exp 3, 4, and 5 present finer edges compared to the binary edge maps predicted in Exp 2, which provides more effective reference for understanding and analyzing relief data.

Moreover, we also observed that Exp 3 was achieved easier convergence on our limited dataset compared to Exp 4 and 5. This underscores the effectiveness of multi-class edge classification and the introduction of the dynamic control parameter $k$. Exp 4 and 5 exhibited poor convergence resulting in sub-optimal error rates, with Exp 4 specifically yielding unsatisfactory extraction results for continuous edge maps as illustrated in Figure 5. Please note that the quantitative results of the edge maps on Exp 2 are for reference only due to the use of different ground truth.

## 4.3 Comparison Results

In this section, we present both quantitative and qualitative comparison results of our method with other state-of-the-art approaches in depth estimation and semantic segmentation tasks as shown in Table 3, Table 4 and Figure 6. More results are provided in our supplementary materials.

For the depth estimation task, we compare our results with soft edge-enhanced network [32], BTS [26], AdaBins [6], DenseDepth [2], Swinmim [55], and the zero-shot model Depth Anything [59]. We observed that, except for the soft edge-enhanced network proposed by Pan et al., which only managed to capture a small portion of the fine structure, other state-of-the-art models designed for public datasets failed to capture the fine structure of the relief, particularly

in the soft edge regions. The zero-shot model, Depth Anything, also failed to generalize in relief scenes, with barely any details extracted. In Figure 6, we illustratively present detailed results from Pan et al., Swinmim, Depth Anything with our results. We achieved the best depth estimation results, capturing the clearest details necessary for forming the fine structure without requiring additional prior information on the test data. The quantitative comparisons presented in Table 3 further substantiate our claims of superior performance compared to other models. Additionally, we reconstruct relief-type cultural heritage into 3D point clouds based on the depth estimation result and calculate the cloud-to-cloud distance with the GT on our validation data.

**Table 3: Compression Results of Depth estimation task.**

| | higher is better | | | lower is better | | | | |
|---|---|---|---|---|---|---|---|---|
| | $\theta_1 \leqslant 1.25$ | $\theta_2 \leqslant 1.25^2$ | $\theta_3 \leqslant 1.25^3$ | RMSE | RMSElog | abs | sq | 3D distance |
| Eigen [13] | 0.306 | 0.598 | 0.778 | 10.289 | 0.781 | 3.067 | 2.001 | 14.982mm |
| Laina [25] | 0.344 | 0.608 | 0.778 | 10.17 | 0.589 | 3.029 | 1.770 | 8.901mm |
| DenseDepth [2] | 0.378 | 0.642 | 0.791 | 9.996 | 0.633 | 3.872 | 2.194 | 7.772mm |
| BTS [26] | 0.441 | 0.770 | 0.921 | 9.841 | 0.455 | 4.074 | 2.128 | 7.897mm |
| AdaBins [6] | 0.340 | 0.648 | 0.828 | 9.870 | 0.562 | 3.519 | 2.117 | 7.691mm |
| Swinmim [55] | 0.360 | 0.730 | 0.875 | 9.942 | 0.456 | 4.073 | 1.969 | 9.902mm |
| DA(zero-shot) [59] | 0.320 | 0.567 | 0.728 | 10.041 | 0.818 | **2.371** | 1.929 | 7.128mm |
| Pan (extra input) [32] | 0.482 | 0.811 | **0.947** | 9.643 | 0.415 | 3.888 | 1.960 | 5.867mm |
| Ours | **0.571** | **0.849** | 0.939 | **9.276** | **0.401** | 3.594 | **1.883** | **5.680mm** |

For the semantic segmentation task, we compare our results with those of JI et al. [17], SegNet [4], U-Net [38], PSPNet [65], DeepLabV3+ [9], EMSANet [41], and EMSAFormer [14]. As illustrated in Table 4, our proposed method outperforms most related works, with the exception of the approach proposed by JI et al. This exception can be attributed to the use of both depth labels and soft edge labels as additional inputs in the model proposed by JI et al., which significantly enhanced the performance of their model. Furthermore, our dataset containing 11 relief panels is smaller than that used by Ji et al., which includes 26 relief panels.

**Table 4: Compression Results of semantic segmentation task.**

| Network | Recall | Precision | mIoU | F1-Score | Accuracy |
|---|---|---|---|---|---|
| SegNet [4] | 0.6843 | 0.7079 | 0.5388 | 0.6932 | 0.7240 |
| U-Net [38] | 0.6708 | 0.7084 | 0.5251 | 0.6839 | 0.7120 |
| PSPNet [65] | 0.6953 | 0.7096 | 0.5470 | 0.7011 | 0.7184 |
| DeeplabV3+ [9] | 0.6707 | 0.6877 | 0.5200 | 0.6777 | 0.6993 |
| EMSANet [14] | 0.8301 | 0.8037 | 0.6980 | 0.8131 | 0.8646 |
| EMSAFormer [14] | 0.7818 | 0.7668 | 0.6479 | 0.7704 | 0.8397 |
| JI (extra input) [17] | **0.8961** | **0.8983** | **0.8158** | **0.8968** | **0.9053** |
| Ours | 0.8385 | 0.8131 | 0.7163 | 0.8243 | 0.8753 |

The proposed method has been applied to the Borobudur temple, extracting multi-modal feature maps, including depth maps, from old photo that recorded the appearance of 156 buried relief panels. This approach has enabled the 3D digital reconstruction of these panels, as illustrated in Figure 1. While there is no ground truth available for quantitative comparisons of the buried relief objects, we provide more qualitative comparison results in our supplementary materials.

## 5 CONCLUSION

In this study, we developed a multi-task learning-based method to predict multiple feature maps containing depth, semantics, and

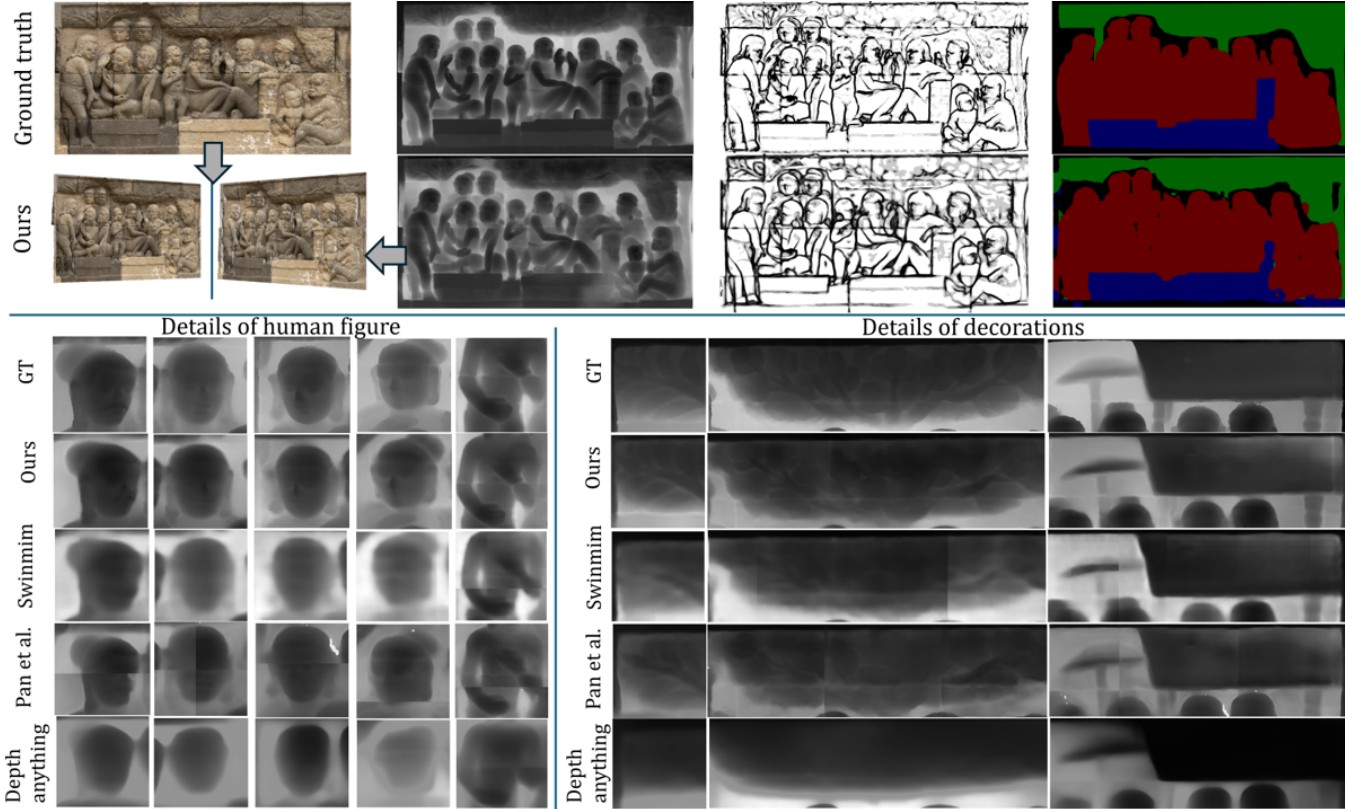

**Figure 6: The top half presents our best experimental results, from left to right: 3D reconstruction model, depth map, soft edge map, and semantic map. The bottom half presents the detailed depth estimation results compared with current SOTA models.**

edges from a single old photo. The proposed method not only provided comprehensive references for multi-modal understanding and analysis but also enabled 3D digital reconstruction of relief-type cultural heritage objects. Through the optimization measures proposed in this paper, the limitations of related studies were effectively solved, and more precise 3D reconstruction models were provided, which were quantitatively and qualitatively verified on the Borobudur dataset. For future work, we may employ a point cloud completion process over the 3D reconstruction model obtained in this work to repair small portions of the side structural information that cannot be covered within a monocular old photo. Furthermore, we aim to extend our method to additional damaged or lost relief-type cultural heritage sites and develop a more comprehensive relief dataset.

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

Received 20 February 2007; revised 12 March 2009; accepted 5 June 2009

