# OpenReview forum: "Reconstructing, Understanding, and Analyzing Relief Type Cultural Heritage from a Single Old Photo"
_acmmm.org/ACMMM/2024/Conference — MM2024 Oral_

### Official Review · Reviewer_SsX2 · 2024-05-14

**Rating:** 4
**Confidence:** 3

**Summary:**

The paper deals with the specific object of relief-type cultural heritage. It proposes a solution for 3d reconstruction of the relief using a single old photograph. The proposed approach apprehends the relief as a 2.5D scene and exploits multiple modalities to tackle the challenge of fine depth structures reconstruction. The proposed method both helps virtually reconstruct potentially damaged elements but also offers a novel basis for analyzing such contents.

**Strengths:**

First of all, the paper is quite clear and pretty well written. The position of the proposed approach within the existing literature. The presentation of the existing limitations and the description of the proposed solutions to each of them are very clear and well thought out.

The proposed idea of exploiting a multi-task approach has merits. Indeed, each output serves a global goal of reconstruction and helps in the soft-edge detection but also ensures that multiple outputs are created, thus increasing the analysis potential and helping understanding these specific contents.
More specifically, the proposed soft-edge detector applied to both the 2D image and the output depth map, ensuring a better detection of soft edges and a more accurate 3D reconstruction overall, is very interesting.

The thought process on the loss used in the model is quite interesting and well presented. The idea of the dynamic control parameter for the loss makes a lot of sense to justify the use of the soft-edge detector module with both ends as input.

The overall evaluation seems to prove the merits and performance of the approach. Visual results in the supplementary material are also quite convincing. Overall, both the scientific interest and the proposed approach are certain.

**Limitations:**

Although the paper is interesting and the proposed approach has merits, some aspects should be addressed in my opinion to improve overall understanding.

First of all, some minor corrections would be beneficial. First, the final sentence of the abstract is both strange and generic, maybe it should be modified or removed. Second, the light grey legend on top of figure 4 is quite unreadable, a bit darker would be better. Third, in section 4.2, mention is made of table 5 rather than table 2 in my opinion…

A second aspect that brought up questions is the classification of the soft-edge map. Although the idea is interesting and the corresponding modified loss is clear, the reason as to why there is only 6 classes and why they are created with such values is unclear. It seems arbitrary and I feel it should be justified.

A third aspect of the paper that should be addressed is the description of the network. The fact that finetuning is not an option is not problematic but I felt that the textual description of the network sometimes does not give more information than Figure 4. It would be interesting to know why such encoders have been used and not others. Textually describing the presence of convolutional layers or skip connections for instance is not informative, especially when the network is fully represented in Figure 4. Instead, more information on which classes are used for the semantic segmentation or insight on whether or not more semantic classes could lead to a better depth map could be beneficial.

Finally, mentions to the supplementary material could be made in several places to help mitigate the impression of lacking information. In my opinion, this could mostly be useful when quickly introducing the dataset (whose creation is well detailed in the Supp. Mat.) and also at the end, when presenting visual results, to guide readers towards the very interesting full-size visual comparisons of the reconstructions.

Typos :
- l. 28 : “high compression” rather than “highly compression” ?
- l. 241, 783, 785, etc… : JI  rather than Ji ?
- l. 334 : strange reference to section 1.

**Suitability:**

2

---

### Official Review · Reviewer_hVh8 · 2024-05-24

**Rating:** 5
**Confidence:** 3

**Summary:**

This paper introduces an innovative method for three-dimensional digital reconstruction, understanding, and analysis of relief-type objects in cultural heritage, based on a single old photograph. Utilizing a multi-task neural network, it predicts multi-modal feature maps that include depth, semantics, and edges, which are crucial for enhancing the information processing capabilities of computer vision tasks. Our network is capable of extracting features ranging from simple geometric details to complex semantic layers, and it adeptly handles edge detection, a task that is often challenging to achieve simultaneously in traditional single-task models.

Moreover, their approach focuses on preserving and restoring damaged or partially lost relief artworks by reconstructing accurate 3D models from archival photographs, thus playing a vital role in the protection and study of cultural heritage. This paper also demonstrates how these reconstructed models can be used for further analyses, such as assessing the degree of material wear and exploring historical transformations, offering new perspectives and tools for comprehensive cultural heritage research.

Their method has been applied to several real-world heritage cases, including the famous reliefs at Borobudur Temple, validating its effectiveness and practicality. Experimental results show significant improvements over current state-of-the-art methods in terms of reconstruction quality, semantic understanding, and edge clarity, thereby confirming its potential contribution to the preservation and study of global cultural heritage.

**Strengths:**

Innovative Approach: This paper leverages an innovative multi-task neural network that predicts multi-modal feature maps including depth, semantics, and edges from a single old photograph. This approach not only enhances traditional image processing capabilities but also integrates multiple complex tasks that are typically handled separately. The ability to simultaneously manage multiple aspects of computer vision within a single framework significantly advances the field, particularly in the context of cultural heritage where detailed and accurate reconstructions are crucial.

Comprehensive Application to Cultural Heritage: The methodology proposed in this paper is particularly beneficial for the cultural heritage field, where preserving the integrity and understanding the context of artifacts is essential. By enabling high-precision 3D reconstructions from merely a single photograph, the method offers a powerful tool for historians, archaeologists, and conservators. These reconstructions can serve multiple purposes, from virtual tourism to detailed analytical studies, thus broadening the scope of how cultural heritage is interacted with and appreciated.

Extensive Experimental Validation: The paper provides extensive empirical evidence of its method's efficacy through its application on a variety of real-world objects, including intricate reliefs from the Borobudur Temple. The experiments compared favorably against current state-of-the-art techniques, showing superior performance in reconstruction accuracy, semantic interpretation, and edge definition. This extensive testing not only substantiates the robustness of the approach but also demonstrates its adaptability to different types of cultural heritage objects and conditions.

Practicality in Preserving Artifacts: The research addresses a critical need in cultural heritage preservation — the ability to accurately reconstruct and analyze ancient artifacts that may no longer be fully intact. This capability is invaluable for restoration efforts and provides a digital means to preserve artifacts' details indefinitely, which is especially important for those at risk of deterioration or further damage.

**Limitations:**

Complexity of Implementation: One significant drawback of the proposed method is its implementation complexity. The multi-task neural network, while robust, requires significant computational resources and expertise in machine learning to deploy effectively. This complexity could limit the accessibility of the technique, especially for institutions with limited technical infrastructure or expertise. Additionally, the complexity might hinder real-time applications or on-site use where computational resources are more constrained.

Dependence on High-Quality Input: The success of the method heavily relies on the quality of the input photograph. While the paper demonstrates impressive results with high-quality images, the performance in scenarios where the available photos are degraded or suboptimal—common issues in archival materials—is not thoroughly addressed. This dependence raises concerns about the method's applicability across a broader range of real-world conditions where old photographs might be faded, blurred, or otherwise impaired.

Limited Generalization Across Diverse Artifacts: Although the paper provides validation through experiments on selected datasets, there is a noticeable gap in demonstrating the method’s effectiveness across a more diverse range of cultural heritage objects. Different artifacts might have varying surface textures, materials, and degradation states, which may affect the model's ability to generalize effectively. The paper could benefit from a broader evaluation that includes a wider variety of artifact types to truly assess the robustness and versatility of the proposed method.

Potential Overfitting and Model Bias: The paper does not sufficiently address the risks associated with overfitting to the specific types of data used in training, particularly given the highly specialized nature of the data involved in cultural heritage. There is also a risk of bias in model predictions due to this overfitting, which could lead to inaccurate reconstructions of less-represented artifact types or styles, potentially skewing historical interpretations.

**Suitability:**

2

---

### Official Review · Reviewer_kYEz · 2024-05-24

**Rating:** 4
**Confidence:** 2

**Summary:**

This paper proposes a multitask model to produce three types of feature maps (depth, semantics and edges) which are then used for better predicting depth in the context of 3D digital reconstructions from 2D images.

The main application of their method is the 3D reconstruction of damaged relief-type cultural heritage objects, whose original versions are often documented in old monocular photos. They explore the prior that such objects are flat on one side, which allows their 3D model to be obtained from a single monocular picture.

The authors tackle the limitations of a previous work on 3D reconstructions from monocular pictures, namely the lack of precision in the estimation of soft edges. This is done a through multitask network, which learns both soft edge detection and depth estimation, combined with a novel edge matching module to focus specifically on soft edges.

Their model was tested on the Borobudur relief dataset, which contains more than six thousand patches of relief objects extracted from eleven high resolution monocular images.

**Strengths:**

The proposed method is well described, and the experimental settings are adequate. The experimental results on the Borobudur dataset show the superiority of the proposed method with respect to eight state-of-the-art depth estimation methods, including a recent approach published in CVPR 2023.

**Limitations:**

In the "Edge detection" subsection, the authors cite some methods for binary edge detection, but other contour detection models which output more than binary edges are not mentioned, such as:

Maninis, Kevis-Kokitsi, et al. "Convolutional oriented boundaries: From image segmentation to high-level tasks." _IEEE transactions on pattern analysis and machine intelligence_ 40.4 (2017): 819-833.

Liu, Yun, et al. "Semantic edge detection with diverse deep supervision." _International Journal of Computer Vision_ 130.1 (2022): 179-198.

Since the extraction of edges and semantics is one of the main contributions of this paper, a broader discussion on other non-binary edge detectors should be included.

Figure 4 contains a lot of information, and some of the text inside it is blurry.

To better evaluate the proposed model against state-of-the-art methods, it would be relevant to include experiments on standard depth datasets, such as Cityscapes and KITTI.

**Suitability:**

2

---

### Meta-Review · Area_Chair_7ca7 · 2024-07-05

**Recommendation:** Accept (Oral)
**Confidence:** 5

**Metareview:**

This study presents a unified framework designed to reconstruct single photos of relief-type cultural heritage objects, addressing challenges such as low resolution and varying degrees of damage and deterioration. The key innovation is a multi-task network that leverages three types of maps—edges, semantics, and depths—together for the reconstruction task. Experiments on public datasets showed impressive results compared to state-of-the-art methods.

The manuscript received three reviews, all recommending acceptance. The chair agrees with the reviewers that the originality of the proposed method meets the standards of ACM MM.